# The First Record of Non-Indigenous Cladoceran *Evadne nordmanni* Lovén, 1836 (Cladocera, Podonidae) in the Middle Part of the Caspian Sea

**Moldir Aubakirova** [1,*], **Elena Krupa** [1], **Igor Magda** [1], **Saule Zh. Assylbekova** [2], **Almat Abayev** [1], **Berdibek Abilov** [2], **Artur Tumenov** [2], **Kuanysh B. Isbekov** [2] **and Zhanara Mazhibayeva** [2]

1  Institute of Zoology, Almaty 050060, Kazakhstan
2  Fisheries Research and Production Center, Almaty 050016, Kazakhstan
*  Correspondence: judo_moldir@mail.ru; Tel.: +7-(272)-694876

**Abstract:** The introduction and spread of non-indigenous species may have ecological, environmental and economic impacts where they invade. This work aims to study the morphological characteristics, the quantitative variables, the possibility of coexistence with other native species and the pathways of introduction of non-indigenous cladoceran *Evadne nordmanni* (Lovén, 1836) in the middle part of the Caspian Sea. Ballast water is a possible vector for the introduction of cladoceran *Evadne nordmanni* into the Caspian Sea. The abundance of *Evadne nordmanni* in all surveyed areas reached an average of 799 individuals/m$^3$. Its biomass was 257.58 mg/m$^3$. *Evadne nordmanni* significantly contributes to the abundance and biomass of zooplankton in the Middle Caspian Sea. The proportion of the dominant calanoida *Acartia tonsa* decreased from 71–90% to 40% with the appearance of *Evadne nordmanni*. Further investigations are needed to analyze the responsible route of *Evadne nordmanni* introduction to the Caspian Sea and its consequences on biodiversity; since this species is a predator and could have consequences on the feeding conditions of planktivorous fish in the Caspian Sea.

**Keywords:** non-indigenous species; native species; interspecific competition; coexistence; forage base; ballast water; zooplankton

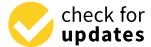



## 1. Introduction

Podonids are predators [1] and typical representatives of freshwater, brackish and marine environments [2]. Ten podonid cladocerans are registered in the Pontocaspian fauna; two *Evadne anonyx* G. Sars and *Evadne prolongata* A. Beaning belong to the genus Evadne [2]. Intercontinental invasions of representatives of Evadne in marine environments have been registered several times [3–5].

The Caspian Sea is one of the primary recipients of non-indigenous species, since it is located on the border of several states and has shipping links and ferry crossings, feeds with transboundary rivers, and functions as a wintering place for migratory birds [6,7]. Over the past 50 years copepoda *Acartia tonsa* [8], *Calanipeda aquaedulcis* Krichagin [9], ctenophores *Mnemiopsis leidyi* (A. Agasis) [10], and *Beroe ovata* Bruguière [11] have invaded the Caspian Sea. The invasion of non-indigenous species has had both positive and negative impacts on aquatic ecosystems of the Caspian Sea. The positive side includes the enrichment of the food base of local species [12]. The appearance of the copepod *Acartia tonsa* Dana and *Calanipeda aquaedulcis* in the Caspian Sea is a vivid example of this, since they belong to the family of calanoid copepods, which have high nutritional value, and contribute to the forage of all fish larvae [13,14]. Representatives of Calanoida family are a favourite object of aquaculture [15]. According to the literature, in 2020, the total production of aquaculture products worldwide reached 214 million tons, 60% more than in 2000 [16]. However, not all invading species have had a positive impact on the ecosystem of the Caspian Sea. Some displaced native species by leading a predatory way of life and creating interspecific

competition for food resources and habitat [17]. For example, native copepods *Eurytemora grimmi* (G. O. Sars, 1897) and *Eurytemora minor* (Behning, 1938) began to disappear with the appearance of ctenophore *Mnemiopsis leidyi* (A. Agasis) in the Caspian Sea [18]. In the Caspian Sea, ctenophore *Mnemiopsis leidyi* had a great negative impact on the planktivorous fish anchovy sprat (*Clupeonella engrauliformes* Borodin, 1904), which is the primary food source of mature sturgeons (family Acipenseridae) and Caspian tulka (genus *Clupeonella*). The anchovy sprat population in the Caspian Sea has not yet recovered [10].

The above reasons prove the need for data on quantitative variables and the possibility of coexistence with the native species of each introduced non-indigenous species. In addition, the problem of occurrence of non-indigenous species becomes especially topical if it appears in a large water body like the Caspian Sea. The Caspian Sea is located at the cross of Europe and Asia. Besides oil and gas production, more than 90% of the world's sturgeon catch is also carried out in this sea, so the economies of coastal countries depend on the Caspian Sea [19]. In this regard, this work aimed to study the morphological characteristics, the quantitative variables, the possibility of coexistence with other species and the possible pathways of introduction of non-indigenous cladoceran *Evadne nordmanni* (Lovén, 1836) in the middle part of the Caspian Sea.

## 2. Materials and Methods

### 2.1. Description of Study Area

The Caspian Sea is the largest inland water body, with an area of 390,000 km$^2$ (Figure 1). According to the physical and geographical conditions, the sea is divided into the Northern, Middle and Southern Caspian. The greatest depth of the middle part of the sea is 788 m. The largest tributaries include The Volga, Ural, Terek, Sulak, and Emba rivers. The salinity of water varies from 12.6 to 13.2%. One hundred and fifty-nine fish species are registered in the Caspian Sea; 28 of them are objects of commercial fishing [20]. The Caspian Sea has shipping links with the Baltic and White Seas through the Volga–Baltic waterway and the White Sea–Baltic Channel [6,19]. The Caspian Sea countries are Kazakhstan, Iran, Turkmenistan, Russia, and Azerbaijan.

### 2.2. Field Sampling

Zooplankton studies of the middle part of the Caspian Sea (Kazakhstan territory) were carried out in May 2021. Planktonic invertebrates of the Middle Caspian Sea were caught from the bottom (deep layer, 97 m) and from the surface (upper layer, 22 m). A total of 16 samples were collected, among them eight samples from the upper layer, 22 m, and eight samples from the deep layer, 97 m. The station coordinates were determined using a GPS navigator (Garmin, Ltd., Olathe, KS, USA) (Table 1). Zooplankton was sampled using a Juday plankton net (mesh size 64 μm) by pulling it from the bottom to the surface. Filtered water was poured into 250 mL plastic bottles and preserved with 40% formalin to a final concentration of 4% [21]. Further processing of the samples was carried out in the laboratory.

**Table 1.** The station coordinates and depths.

| Station | Depth, m | Coordinates | |
| :---: | :---: | :---: | :---: |
| | | Longitude | Latitude |
| 1 | 97 | | |
| 2 | 22 | 51°48.3535′ E | 42°07.2959′ N |
| 3 | 97 | | |
| 4 | 22 | 51°48.3535′ E | 42°07.5659′ N |
| 5 | 97 | | |
| 6 | 22 | 51°48.3535′ E | 42°07.836′ N |
| 7 | 97 | | |
| 8 | 22 | 51°47.6347′ E | 42°08.106′ N |

**Table 1.** *Cont.*

| Station | Depth, m | Coordinates | |
| --- | --- | --- | --- |
| | | Longitude | Latitude |
| 9 | 97 | 51°47.9941′ E | 42°08.106′ N |
| 10 | 22 | | |
| 11 | 97 | 51°44.7594′ E | 42°10.8065′ N |
| 12 | 22 | | |
| 13 | 97 | 51°51.9476′ E | 42°10.8065′ N |
| 14 | 22 | | |
| 15 | 97 | 51°51.9476′ E | 42°05.4055′ N |
| 16 | 22 | | |

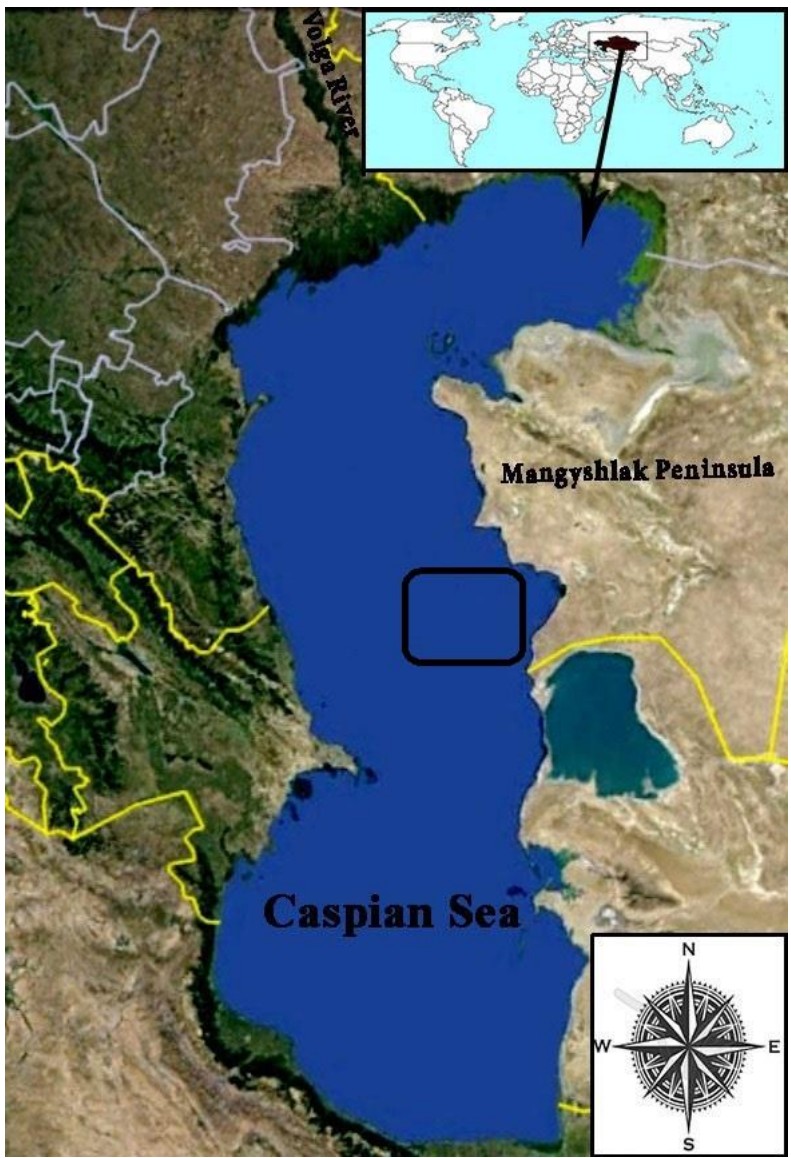

**Figure 1.** Map-scheme of the location of the sampling station (black square) in the middle part of the Caspian Sea, May 2021.

### 2.3. Laboratory Processing

Planktonic invertebrates were identified at the species level using the determinants of the respective groups and genera [9,22–25]. Quantitative sample processing was carried

out by standard methods [21]. Planktonic invertebrates were counted in a particular part of the sample. Counting of planktonic invertebrates started by concentrating the sample at 300 cm$^3$. After thorough mixing, three parts of the sample were taken using a Stempel pipette with a volume of 1 mL. Different age stages of species (the most numerous) encountered in this subsample were counted in the Bogorov counting chamber. Next, the sample was concentrated to half of the previous volume. Three sub-samples were taken from it, in which non abundant species was counted. The process was repeated when the sample was concentrated to 25 cm$^3$. The number of individuals of rare species was counted by viewing the entire sample. Adult females, females with eggs, males, copepodites at life stages 1–3 and 4–5 and nauplii were counted and measured separately in copepods. In cladocerans, females with juveniles in the brood pouch, sterile females, males, and juveniles were counted separately. A calculation of an average individual mass of a specimen was performed as the total biomass divided by the total abundance of zooplankton [21]. During the calculation of the individual mass of planktonic invertebrates, the formula of dependence between mass and body length was used [26].

For each species, the total abundance and biomass were calculated. The obtained results were recalculated per 1 m$^3$.

$$N = \frac{n \times \left(\frac{V_1}{V_2}\right)}{V_3} \tag{1}$$

where N represents abundance (individual/m$^3$), n, the number of individuals per parts (individual), $V_1$, the concentration volume (cm$^3$), $V_2$, the subsample volume (cm$^3$), and $V_3$, the volume of filtered water (m$^3$).

The volume of filtered water was calculated by the formula:

$$V_3 = h \times \pi r^2 \tag{2}$$

where $V_3$ represents the volume of filtered water, h, the depth of caught water column, $\pi$, the mathematical constant ($\pi \approx 3.14$), and r, the internal radius of the inlet hole of the Juday plankton net.

Identification of dominant species was carried out according to the Lyubarsky's scale [27]. According to this scale, a list of absolute dominants includes the species that created more than 60% of the quantitative variables of the community. The species that made up more than 31–60% of the quantitative variables of the community were included in the list of dominants. Subdominants included species that contributed 10–30% to quantitative variables of the community.

### 2.4. Statistical Analysis and Comparisons with Previous Studies

We used statistical methods for ease of perception of information about the distribution of *Evadne nordmanni* in different depths. Boxplots were created in R by using the boxplot function [28].

The obtained data were compared with studies from previous years. Field sampling (stations, seasons, and depths) and laboratory processing (species identification and quantification methods) of the studies in 2020 [29] were consistent with the current sampling and methods. All methods of the 2008 studies, except for sample collection, were the same as the current sampling and methods [30]. Samples were collected only from the surface (upper layer, 38.4 m) in 2008 [30].

## 3. Results

### 3.1. Morphological Characteristics of Evadne nordmanni

An important morphological characteristic of *Evadne nordmanni* that distinguishes it from other species of the genus *Evadne* is the presence of one setae on the exopodite of the third thoracic limb [24] (Figure 2d). The setae formula of exopodites of I–IV thoracic limbs

is 2.2.1.1. The head does not separate from the shell. The swimming antennules are small and the apical segments are tiny. Exopodites in thoracic limbs (I–IV) are very short. The maxillary outgrowth is well developed on the second and third thoracic limbs and includes two large teeth; on the first thoracic limb, it is in the shape of a small tubercle with setae. Apical setae on the endopodite of the first thoracic limb are long and thin. Apical setae on endopodite of the II and IV thoracic limbs are short and have the shape of a claw [24].

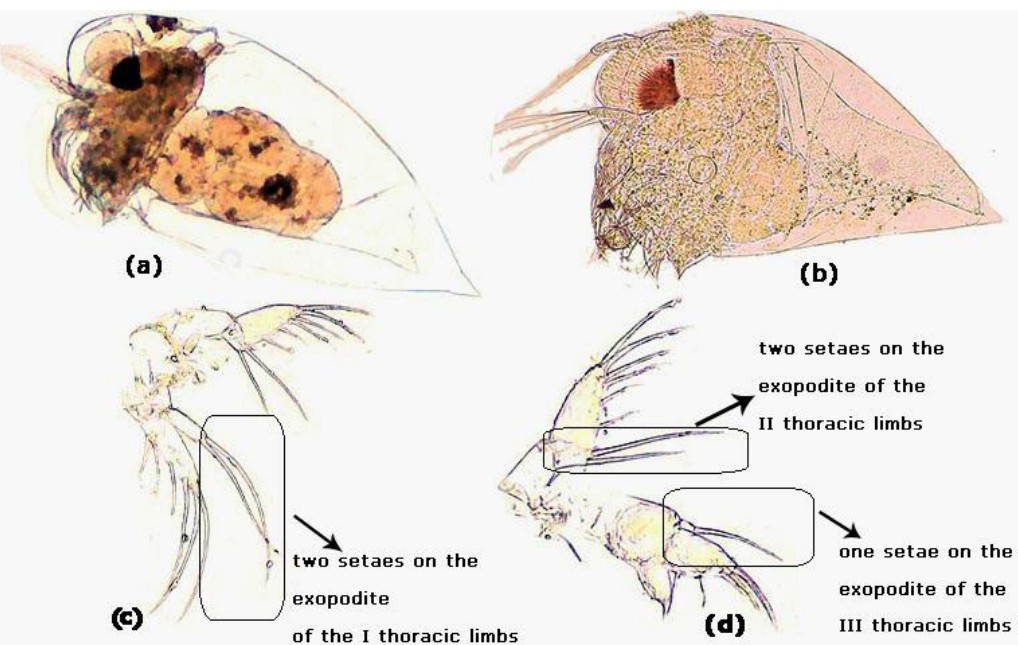

**Figure 2.** General form and I, II, III thoracic limbs of *Evadne nordmanni.* (**a**) general form of female; (**b**) general form of male; (**c**) I and II thoracic limbs; (**d**) II, III thoracic limbs.

The females and males of cladoceran *Evadne nordmanni* examined in our samples had different body lengths (Figure 2). The body length of the females varied from 0.7 to 0.9 mm and the shell height ranged from 1.2 to 1.4 mm. The shell was elongated with a slightly pointed end. Adult females had several embryos in the brood pouch (Figure 2a). The length of males was within 0.60–0.63 mm, and shell height was 0.7–0.8 mm. The shell is triangular, slightly elongated, with a somewhat pointed end. The penis is long, thin, and rounded (Figure 2b).

### 3.2. *Abundance and Biomass of Evadne nordmanni*

The abundance of *Evadne nordmanni* in all surveyed areas reached an average of 799 individuals/m³. The biomass was 257.58 mg/m³ (Figure 3). According to the boxplots, cladocerans were widespread in the water column (deep layer, 97 m and upper one, 22 m), as abundance and biomass of the species were equal at both depths.

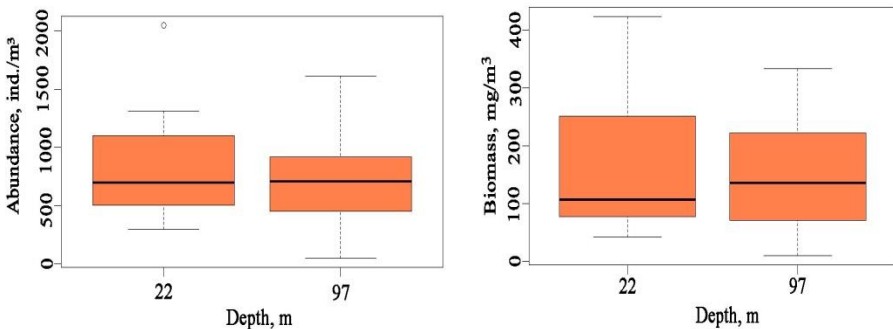

**Figure 3.** Quantitative variables of *Evadne nordmanni* in different depths of the Middle Caspian Sea.

### 3.3. Coexistence of Evadne nordmanni with Other Planktonic Invertebrates in the Middle Caspian Sea

A total of 11 more taxa of zooplankton were identified along with the cladoceran *Evadne nordmanni* in the Middle Caspian Sea. Cladocerans *Evadne nordmanni*, *Evadne anonyx* (G. Sars), *Pleopis polyphemoides* (Leukart), copepods *Acartia tonsa* (Dana), and temporary inhabitants of the water column, such as larvae of Cirripedia, Ostracoda, Bivalvia, were widespread in the surveyed water area of the Middle Caspian Sea.

The abundance of planktonic invertebrates reached only 4364 individuals/m$^3$ with biomass of 187.43 mg/m$^3$ (Table 2). The copepods *Acartia tonsa*, cladocerans *Evadne nordmanni*, and larvae of Cirripedia dominated. The absolute dominance in zooplankton biomass was the cladoceran *Evadne nordmanni*, which contributed 84.9% to the total biomass of the community.

**Table 2.** Quantitative variables of zooplankton communities in the Middle Caspian Sea, May 2021 (average values with standard deviation).

| Taxa | Abundance | Contribution % | Biomass | Contribution % |
|---|---|---|---|---|
| Rotifera | | | | |
| *Synchaeta littoralis* Rousselet, 1902 | 2.8 ± 2.7 | 0.06 | 0.01 ± 0.01 | 0.006 |
| Cladocera | | | | |
| *Evadne anonyx* G.O. Sars, 1897 | 2.7 ± 0.9 | 0.06 | 0.2 ± 0.1 | 0.117 |
| *Pleopis polyphemoides* Leuckart, 1859 | 26.7 ± 14.9 | 0.61 | 2.6 ± 1.7 | 1.390 |
| *Podon intermedius* Lilljeborg, 1853 | 13.7 ± 11.3 | 0.31 | 1.5 ± 1.3 | 0.817 |
| *Evadne nordmanni* Lovén, 1836 | 799.4 ± 127.6 | 18.30 | 159.1 ± 28.8 | 84.891 |
| Copepoda | | | | |
| *Acartia tonsa* Dana, 1849 | 1746.8 ± 351.7 | 40.0 | 13.6 ± 3.0 | 7.275 |
| *Calanipeda aquaedulcis* Krichagin, 1873 | 0.2 ± 0.1 | 0.004 | 0.0003 ± 0.0003 | 0.0002 |
| *Halicyclops sarsi* Akatova, 1935 | 0.8 ± 0.5 | 0.02 | 0.002 ± 0.001 | 0.001 |
| Others | | | | |
| *Bivalvia* gen.sp. | 21.6 ± 4.3 | 0.49 | 0.09 ± 0.02 | 0.046 |
| *Spionidae* sp. | 0.7 ± 0.4 | 0.02 | 0.003 ± 0.003 | 0.002 |
| *Cirripedia* gen.sp. | 1457.3 ± 356.8 | 33.4 | 2.9 ± 0.7 | 1.573 |
| *Ostracoda* gen.sp. | 291.4 ± 58.7 | 6.7 | 7.3 ± 1.5 | 3.880 |
| Total | 4364.1 ± 751.1 | 100 | 187.4 ± 30.6 | 100 |

## 4. Discussion

### 4.1. Pathways of Introduction of Evadne nordmanni to the Caspian Sea

Cladoceran *Evadne nordmanni* is native to the Baltic Sea [4]. It is mainly distributed in the Atlantic and Pacific Oceans and adjacent seas, such as the White Sea, Baltic Sea, Mediterranean Sea, and Black Sea [25]. *Evadne nordmanni* was not recorded in any part of the Caspian Sea until 2021 [24,29–31]. It is well known that aquaculture activity, transboundary rivers, migratory birds [32] and ballast water are the main reasons for the intercontinental invasions of aquatic species [33,34]. Migratory birds could be a potential vector of *Evadne nordmanni* introduction in the Caspian Sea since five species of waterfowl arrive from Europe for wintering [7]. In addition, the biological characteristic of cladoceran *Evadne nordmanni* whereby it produces resting eggs [24] increases the possibility of changing habitat by means of migratory birds. However, the introduction of non-indigenous aquatic organisms with migratory birds has not been registered for the water bodies of Kazakhstan, and worldwide the number of confirmed cases is low, only 14 [35].

Non-indigenous aquatic species often enter waterbodies of Kazakhstan via transboundary rivers, due to aquaculture activity and ballast water [10,18,36,37]. For example, during the transportation of commercial fish species, grass carp (genus *Ctenopharyngodon*), silver carp (genus *Hypophthalmichthys*) and carp (genus *Cyprinus*) from the Far East and China, freshwater shrimps *Exopalaemon modestus* (Heller, 1862) and *Macrobrachium nipponensis* (De Haan, 1840) invaded the water bodies of South Kazakhstan [36]. The sea crab *Eriocheir sinensis* (H. Milne Edwards, 1853), which originates from China, was intro-

duced into the water bodies of East Kazakhstan through the transboundary Black Irtysh River [37]. However, it is important to note that aquaculture activity and transboundary rivers should not be the reason for the appearance of cladoceran *Evadne nordmanni* in the Caspian Sea. This hypothesis is supported by the fact that large-scale fish stocking works in Kazakhstan have not been carried out since 2000 [38]. Similarly, cladoceran *Evadne nordmanni* also could not have been introduced through the transboundary Volga River, the main tributary of the Caspian Sea, since, at a salinity of less than 12‰, the species discontinues its vital activity [25]. Hence, ballast water is proposed as the possible vector for cladoceran *Evadne nordmanni* introduction into the Caspian Sea. The Caspian Sea has shipping links with the Baltic and White Seas through the Volga–Baltic waterway and the White Sea–Baltic Channel [6,19]. Copepoda *Acartia tonsa* [8], *Calanipeda aquaedulcis* Krichagin [9], ctenophores *Mnemiopsis leidyi* (A. Agasis) [10], *Beroe ovata* Bruguière [11] invaded the Caspian Sea through these routes.

### *4.2. Coexistence of Evadne nordmanni with Other Planktonic Invertebrates in the Middle Caspian Sea and Potential Consequences on Biodiversity*

The species composition of zooplankton, quantitative variables and group of dominant species of the Middle Caspian Sea changed in 2021 compared with previous studies [29,30]. The number of zooplankton taxa in the Middle Caspian Sea decreased from 21 in 2008 to 10–12 in 2020–2021 [29,30] (Table 3). Rotifers *Brachionus quadridentatus* Hermann, 1783, *Synchaeta cecilia* Rousselet, 1902, *Synchaeta stylata* Wierzejski, 1893 and cladocerans *Cornigerius maeoticus hircus* (G.O. Sars, 1902), copepods *Idyaea furcata* (Baird, 1837), *Ergasilidae* gen.sp., *Calanoida* gen.sp. and larvae of *Hediste diversicolor* (O.F. Müller, 1776), Nematoda, recorded in 2008, fell out of the composition of zooplankton of the Middle Caspian Sea in 2021. The species richness of the Middle Caspian Sea zooplankton in 2021 increased relative to 2020, due to the appearance of the cladoceran *Evadne nordmanni* and larvae of Spionidae.

Along with altering the species composition, zooplankton quantitative variables also changed in 2021. The abundance of the community reached 4364 individuals/m$^3$ in 2021. The variable was slightly higher than in previous years [29,30], as in 2008, it was 3800 individuals/m$^3$, and in 2020 abundance reached only 3051 individuals/m$^3$ [29,30]. Biomass of the zooplankton community was equal to 187.43 mg/m$^3$ in 2021, which was higher than the biomass of the community (20.0 mg/m$^3$) in 2008 and lower than the biomass of the community (333.4 mg/m$^3$) in 2020 [29,30]. The higher zooplankton biomass in 2020 is related to the presence of the large-sized ctenophore *Mnemiopsis leidyi* [29] in the zooplankton community.

The composition of the dominant zooplankton species in the surveyed area changed in 2021 compared to data of previous years [29,30]. The complex of dominant species was replenished with the cladoceran *Evadne nordmanni*. The proportion of the permanent dominant calanoida *Acartia tonsa* decreased from 71–90% in 2008 and 2020 to 40% in 2021 [29,30]. In terms of biomass, the dominants of previous years, such as *Acartia tonsa*, *Evadne anonyx* [30] and *Podonevadne camptonyx* [29], were replaced by the cladoceran *Evadne nordmanni*.

Competition for food resources and consumption by *Evadne nordmanni* should be the reason for the decline in abundance and biomass of the dominant copepod *Acartia tonsa*. The food resources of cladoceran *Evadne nordmanni* and copepod *Acartia tonsa*, have similar components. The food base of these species includes diatoms, dinoflagellates and peredinum [23,39–41]. It is possible that by 2021, due to the scarcity of the forage base, *Evadne nordmanni* started to prey on copepodites and nauplii of *Acartia tonsa*. This is supported by the predominance of copepod eggs in the food items of *Evadne nordmanni* in water bodies of Scotland [39], as well as the fact that *Evadne nordmanni* only consumes food of animal origin in the Mediterranean Sea [41].

**Table 3.** The taxonomic changes in zooplankton communities of the Middle Caspian Sea.

| Taxon Name | Years | | |
|---|---|---|---|
| | 2008 [30] | 2020 [29] | 2021 |
| Rotifera | | | |
| *Brachionus quadridentatus* Hermann, 1783 | + | − | − |
| *Synchaeta cecilia* Rousselet, 1902 | + | − | − |
| *Synchaeta littoralis* Rousselet, 1902 | + | − | + |
| *Synchaeta stylata* Wierzejski, 1893 | + | − | − |
| Cladocera | | | |
| *Podonevadne camptonyx* (G.O. Sars, 1897) | + | + | + |
| *Podonevadne angusta* (G.O. Sars, 1902) | + | + | − |
| *Podonevadne trigona* (G.O. Sars, 1897) | + | + | − |
| *Pleopis polyphemoides* (Leuckart, 1859) | + | + | + |
| *Podon intermedius* Lilljeborg, 1853 | + | + | + |
| *Evadne anonyx* Sars,1897 | + | + | + |
| *Cornigerius maeoticus hircus* (GO Sars, 1902) | + | − | − |
| *Evadne nordmanni* Lovén, 1836 | − | − | + |
| Copepoda | | | |
| *Acartia tonsa* Dana, 1849 | + | + | + |
| *Calanipeda aquedulcis* Krichagin, 1873 | + | − | + |
| *Halicyclops sarsi* Akatova, 1935 | + | − | + |
| *Idyaea furcata* (Baird, 1837) | + | − | − |
| *Ergasilidae* gen.sp. | + | − | − |
| *Calanoida* gen.sp. | + | − | − |
| Others | | | |
| *Mnemiopsis leidyi* A. Agassiz, 1865 | − | + | − |
| *Hediste diversicolor* (O.F. Müller, 1776) | + | − | − |
| *Spionidae* sp. | − | − | + |
| *Bivalvia* gen.sp. | + | + | + |
| *Cirripedia* gen.sp. | + | + | + |
| *Nematoda* gen.sp. | + | − | − |
| Total | 21 | 10 | 12 |

The larvae of marine fish species prefer copepods as food rather than cladocerans [15,42]. In this regard, the occurrence of *Evadne nordmanni* and a reducing density and biomass of copepod *Acartia tonsa* could have consequences on the feeding conditions of planktivorous fish in the Caspian Sea and, consequently, their structural variables.

## 5. Conclusions

For the first time, our research revealed the introduction of the cladoceran *Evadne nordmanni* into the Middle Caspian Sea. The most probable route of its appearance is by ballast water. *Evadne nordmanni* significantly contributes to creating zooplankton quantitative variables. The proportion of the dominant calanoida *Acartia tonsa* decreased from 71–90% to 40% with the appearance of *Evadne nordmanni.* Further investigations are needed to analyze the responsible route of *Evadne nordmanni* introduction to the Caspian Sea and its consequences on biodiversity.

**Author Contributions:** Conceptualization, M.A.; methodology, M.A. and E.K.; software, M.A.; validation, M.A.; formal analysis, M.A.; investigation, M.A.; resources, M.A.; data curation, M.A.; writing—original draft preparation, M.A.; writing—review and editing, M.A.; visualization, I.M, S.Z.A., A.A., B.A., A.T., K.B.I. and Z.M.; supervision, M.A.; project administration, I.M.; funding acquisition, I.M. All authors have read and agreed to the published version of the manuscript.

**Funding:** This research was funded by the Institute of Zoology of Ministry of Education and Science of the Republic of Kazakhstan, the Program number OR11465437—"Development of a national electronic data bank on the scientific zoological collection of the Republic of Kazakhstan, ensuring their effective use in science and education" (2021–2022).

**Institutional Review Board Statement:** Not applicable.

**Informed Consent Statement:** Not applicable.

**Data Availability Statement:** No new data were created or analyzed in this study. Data sharing is not applicable to this article.

**Conflicts of Interest:** The authors declare no conflict of interest.

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
