# Peer review of "The First Record of Non-Indigenous Cladoceran Evadne nordmanni Lovén, 1836 (Cladocera, Podonidae) in the Middle Part of the Caspian Sea"

_water, doi:10.3390/w14182818_

Round 1

Reviewer 1 Report

Comments on the manuscript (water-1876561)

General comment:

This manuscript reports the occurrence of the non-indigenous Cladocera in the middle part of the Caspian Sea.  The provided data is limited, and far from a standard study in this field.  While the authors speculated that the cladoceran species would be intruded on by the ballast water of the ship, there is no evidence is provided to confirm this.  It is just speculation.  From this point of view, I want to recommend the rejection of the manuscript for publication.  Since the presentation styles and provided data are still far from the standard paper in this field if the editor tries to accept the manuscript needs to revise for such issues. 

Detailed comments:

L3: Spelling correct “Caspian”.

L70-71: Evadne nordmanni.  While this species is available even for the title, this manuscript is not mentioning of this species at the end of the Introduction.  It is anomalous.  Please provide some notes on the cladoreans and Evadne nordmanni in the earlier part of the introduction. 

L85-93: 2.2. Field Sampling.  Are there any casts of CTD?  Such environmental data may be helpful and valuable to evaluate the environmental effects on the zooplankton community. 

L85-93: 22 m and 97 m.  Why do the authors apply such abrupt two depths?  Are there any reasons to apply such two depths?  I read that the authors collected zooplankton samples from two depths (0-22 m and 0-97 m) at each station.  Is it correct?  It is hard to understand why they applied such two depths and what to show from these two depths. 

L128: over different years.  Are the sampling designs (stations, seasons, depths, and quantification methods) the same for the other years?  If the authors used such data for analysis, providing such information is required to include in the Materials and methods. 

L134-143: It is hard to understand the description of the morphology of Evadne.  From Fig. 2, the readers may not understand where to be mentioned and where to see noticed.  As my idea, please provide names or arrows to indicate points in Figure 2 that may be helpful for the readers. 

L153: one space is needed between ; and (d). 

L157-160: From Fig. 3, I could not see the decrease of values from 0-22 m to 0-97 m.  Is there such decreasing (45.5% in abundance and 47.5% in biomass) available?  Somewhat mistakes may be included. 

L157 and Fig. 3: I could not find mentioning the quantitative methods of biomass in Material and methods. 

L171-176: Again, how to estimate biomass?  Mentioning methods may be needed.  The contrasting contribution in abundance and biomass of Evadne nordmanni and Acartia tonsa suggest that the individual mass is much larger for E. nordmanni than that of A. tonsa.  Is it true?  If so, providing such information (individual mass = biomass/abundance) may also be interesting. 

L181: Baltic Sea.  Are there any cargo vessels connecting the Baltic Sea and the Caspian Sea? 

L213-214: It is just speculation.  It may need more direct evidence to say such a thing.  Have you examined the ballast water of the cargo vessel? 

L240-248: Information on the taxonomic changes between the years (2008, 2020, and 2021) are not obtained from Fig. 4.  The panels showing taxonomic accounts may be better to replace Fig. 4. 

Figures and tables:

Fig. 1: Information on latitude and longitude and horizontal scales should be added.  If possible, providing depth contour information is informative for the readers.  Are there any ports that induce the ballast water by ship cargo?  Providing such information may be valuable for the readers.

Fig. 2: As specific morphological characteristics of the species, indicating names or arrows in the pictures may help the readers to know where is the point.  Providing scale bars may be helpful. 

Fig. 3: It is hard to understand what the authors try to show from this figure. 

Table 1: The sum of the percentage values in this table is over 100%.  It might be due to the four categories at the end of the table (Total Rotifera, Total Cladocera, Total Copepoda, Total Others).  This taxonomic information may be better presented before species, and not providing values (abundance, biomass, and contribution) may be a standard presentation style.  Units may be needed for both abundance and biomass. 

Fig. 4: It is hard to understand what the authors want to show from this figure.  Providing information on the taxonomic accounts each year may be valuable.  

Author Response

Dear Reviewer,

We express our deep gratitude for your comments and suggestions regarding our manuscript. The manuscript has been substantially revised based on your suggestions and comments.

Kind Regards,

Moldir Aubakirova, Elena Krupa, Igor Magda, Saule Assylbekova, Almat Abayev, Berdibek Abilov, Artur Tumenov, Kuanysh Isbekov, Zhanara Mazhibayeva

Reviewer 2 Report

water-1876561

The first record of non-indigenous cladoceran Evadne nordmanni Lovén, 1836 in the middle part of the Caspean Sea

M Aubakirova, et al.

This manuscript presents a new cladoceran occurrence in the Caspian Sea and some possible consequences on the pelagic system. In general, the work is warily designed and written. The major flaw is the last part of the Discussion, which needs to be rewritten. Check carefully the English in the entire manuscript.

TITLE

I strongly suggest using “Caspian” instead of “Caspean”. You are using both throughout text.

ABSTRACT

L25-26 Delete "Our results again confirmed that ballast water is the main introduction pattern for non-indigenous marine species." There is not a confirmation, just an indication, already included above.

MATERIALS AND METHODS

L87-89. Not clear, please rephrase.

RESULTS

L173-175 Replace "The copepods Acartia tonsa, cladocerans Evadne nordmanni, and temporary inhabitants of the water column – larvae of Cirripedia dominated in abundance." with "The copepod Acartia tonsa, cladoceran Evadne nordmanni, and  larvae of Cirripedia dominated."

DISCUSSION

L 181 Replace "Evadne nordmanni is" with "It is"

L 187 Replace  "...can be considered a vector .." with "..could be potentially a vector.."

L190-191 Delete "despite the above facts,"

L203 Replace "..are not the reason .." with "..should not be the reason .."

L204 Replace "This conclusion  .." with "This hypothesis  .."

L208 Replace "Ballast water may be the ..." with "Hence, ballast water is proposed as the ..."

L213 Replace "assumed" with "hypothesized". Please rephrase the whole sentence.

L216 Replace ".. and its possible harm to local biodiversity ..." with " ..and potential consequences on biodiversity..."

L222 " ...from 22 in 2008 to 10-12 in 2020-2021" a citation is needed

L229 Figure 4: The authors should present info (preferably in the materials & methods section) about the data source for the years 2008, 2020.

L252-254 To be rephrased.

L249 - 264 This paragraph should be rewritten, there are several unclear points and editorial errors. Also, delete the comment about juveniles of Acartia tonsa ie “The juveniles of Acartia tonsa ….. presence of ctenophore [21]", the reasoning is poor.

L261-262 Replace "This conclusion is confirmed ..." with "This is supported ....".

L267 Replace "..can affect .." with "..could have consequences.. ".

Author Response

Dear Reviewer,

We express our deep gratitude for your comments and suggestions regarding our manuscript. The manuscript has been substantially revised based on your suggestions and comments.

Kind Regards,

Moldir Aubakirova, Elena Krupa, Igor Magda, Saule Assylbekova, Almat Abayev, Berdibek Abilov, Artur Tumenov, Kuanysh Isbekov, Zhanara Mazhibayeva

Dear Reviewer,

We express our deep gratitude for your comments and suggestions regarding our manuscript. The manuscript has been substantially revised based on your suggestions and comments.

Kind Regards,

Moldir Aubakirova, Elena Krupa, Igor Magda, Saule Assylbekova, Almat Abayev, Berdibek Abilov, Artur Tumenov, Kuanysh Isbekov, Zhanara Mazhibayeva

Dear Reviewer,

We express our deep gratitude for your comments and suggestions regarding our manuscript. The manuscript has been substantially revised based on your suggestions and comments.

Kind Regards,

Moldir Aubakirova, Elena Krupa, Igor Magda, Saule Assylbekova, Almat Abayev, Berdibek Abilov, Artur Tumenov, Kuanysh Isbekov, Zhanara Mazhibayeva

Reviewer 3 Report

I read this paper with interest. I have some suggestions for improvement.

Line 12, do not use “penetration”, instead use “introduction” or “invasion”.

Line 13, the word “them” is ambiguous here. Clarify.

Line 14 “established successfully” is problematic and value-laden wording, please replace.

Line 21, does “permanent” mean “native”?

The abstract needs work to make the meaning clear. I have listed some of the wording problems above, but not all.

Line 32 and throughout, do not use “penetration” or “penetrated”

Figure 1, map should show location within the hemisphere as well as the more localized information.

Line 129, explain the network analysis in detail. And do you have a citation for JASP?

Line 150, include reference to Figure 2 b.

Line 152, do not use the word “habitus” rather use “form”

Lines 157-160, I don’t think this statement is true, the figure shows equal abundance and biomass at both depths. To ays something is different it must have a corresponding statistical test.

Figure 3, the axis labels are too small to read.

Line 171-172, what does it mean that “Quantitative variables… were low”? Compared to what? Where is the statistical basis for calling them low?

Line 180-214, this whole section is relatively weak in its arguments. I agree that ballast water may be most likely, but the truith is you don’t know how it got there. Revise this section to suggest that the most probable route is by ballast water. Don’t say “it can be assumed”.

Figure 4 is not very useful. You must explain how you got this plot and what such an analysis shows in the methods.

Line 232-238, once again, where are the analyses to support this statement?

Line 240-248, if you are going to make quantitative comparisons to previous studies, you must provide greater detail about the previous sampling and show trhat the sampling is consistent with the current sampling and methods.

Lines 249-264, you cannot infer competition without more data. You sampled at only one time, so you cannot describe potential changes through time.

Your conclusions are tenuous at best based on the data you present. Your conclusions should reflect the quality of your data and should not presume you know more than what can be inferred from the scope of your sampling.

Author Response

(The authors gave the same response as above.)

Round 2

Reviewer 1 Report

I think that the authors revised the manuscript sufficiently.  I want to recommend the manuscript be accepted for the journal.  

Reviewer 3 Report

Good work on responding to my suggestions. Some moderate editing for English language usage is still required. I am assuming that the old figure 4 has been deleted, but it still shows up in the text.